# Amplified seasonal cycle in hydroclimate over the Amazon river basin and its plume region

Yu-Chiao Liang [1✉], Min-Hui Lo [2], Chia-Wei Lan [2], Hyodae Seo [1], Caroline C. Ummenhofer [1], Stephen Yeager [3], Ren-Jie Wu[2] & John D. Steffen[1]

The Amazon river basin receives ~2000 mm of precipitation annually and contributes ~17% of global river freshwater input to the oceans; its hydroclimatic variations can exert profound impacts on the marine ecosystem in the Amazon plume region (APR) and have potential far-reaching influences on hydroclimate over the tropical Atlantic. Here, we show that an amplified seasonal cycle of Amazonia precipitation, represented by the annual difference between maximum and minimum values, during the period 1979–2018, leads to enhanced seasonalities in both Amazon river discharge and APR ocean salinity. An atmospheric moisture budget analysis shows that these enhanced seasonal cycles are associated with similar amplifications in the atmospheric vertical and horizontal moisture advections. Hierarchical sensitivity experiments using global climate models quantify the relationships of these enhanced seasonalities. The results suggest that an intensified hydroclimatological cycle may develop in the Amazonia atmosphere-land-ocean coupled system, favouring more extreme terrestrial and marine conditions.

[1] Woods Hole Oceanographic Institution, Woods Hole, MA, USA. [2] Department of Atmospheric Sciences, National Taiwan University, Taipei, Taiwan. [3] National Center for Atmospheric Research, Boulder, CO, USA. ✉email: yliang@whoi.edu

The Amazon river basin (delineated by the black contour in Fig. 1a) receives ~2000 mm of rainfall annually[1]. This vast amount of precipitation feeds the Amazon river, ranked as the world's largest river in terms of annual discharge[2]. The Amazon river discharge contributes ~17% of global river freshwater input to the ocean[2], significantly affecting the physical and biogeochemical upper ocean properties in the coastal and neighboring oceans[3–7]. One prominent feature is the so-called Amazon plume region (APR), which is characterized by relatively low ocean salinity[3,4,8] and high nutrients brought by the Amazon river discharge[3,4,9,10]. Observational evidence suggests that high nutrient contents from the Amazon river discharge help sustain high marine productivity in the APR, with the maximum chlorophyll content concentrated in the upper 5-m ocean[3]. In addition, the mixing of supersaturated Amazon freshwater with undersaturated surface ocean water results in a net sink of atmospheric carbon dioxide within the APR[11,12]. Therefore, the variability of Amazon river discharge can impact marine biogeochemistry, productivity, and the carbon cycle within the APR and surrounding areas[13].

The low salinity water in the APR also significantly increases the upper ocean stratification, creating a thick barrier layer that inhibits the mixing of cold thermocline water into the surface waters[12]. As a consequence of reduced mixing, more heat is trapped in the upper ocean[4,8,14–16]. Due to the APR's areal extent spanning from the Amazon river mouth near the Equator to the East Caribbean Sea (red box in Fig. 1a, see "Methods" for the box definition), the enhanced near-surface heat storage in the APR associated with the barrier layer dynamics provides favorable surface conditions for hurricane genesis over the broad regions of the tropical western Atlantic[8,15]. Moreover, the variability of Amazon freshwater and resultant ocean salinity changes have been suggested to affect tropical Atlantic air–sea interactions[17,18] and the variability of the Atlantic intertropical convergence zone (ITCZ)[19], regional sea-level height changes[20–22], as well as to exert potentially far-reaching impacts on the Atlantic meridional overturning circulation (AMOC)[23]. Thus, understanding the mechanisms for changes in the Amazon river discharge and the associated upper ocean stratification in the APR is important not only for the Atlantic hurricane forecasts but also for improved understanding of basin-scale climate variability.

Recent studies have found that the Amazonia hydro-climatological cycle, manifested as the seasonality changes in precipitation and Amazon river discharge, has become intensified

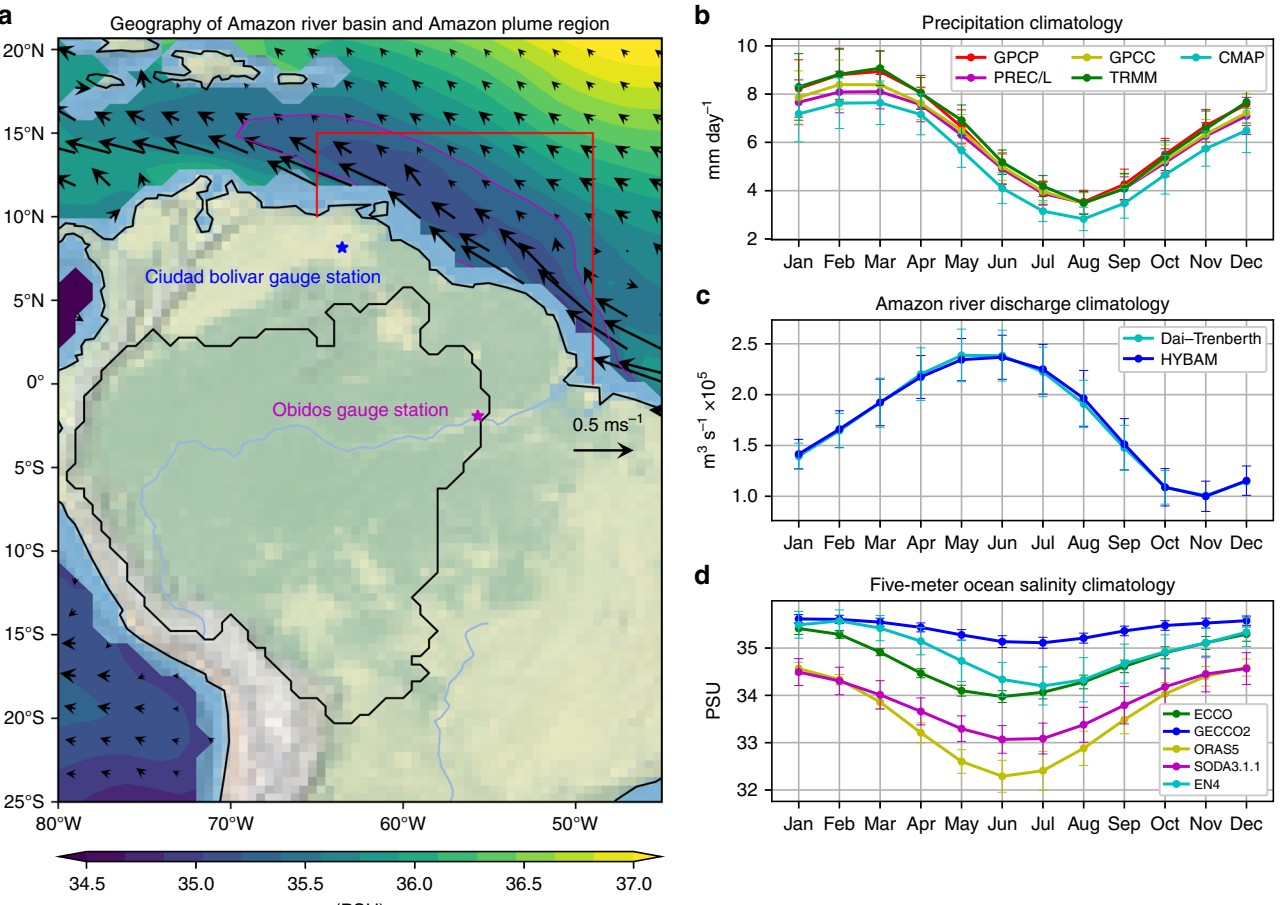

**Fig. 1 Relationships between the seasonal cycles in Amazonia precipitation, Amazon river discharge, and Amazon plume region (APR) ocean salinity.** **a** The geographic domain of Amazon river basin (black contour line) and APR (red box). The color shading over the ocean represents annual mean 5-m ocean salinity with 34.5 PSU countered as magenta, and the black arrows denote annual mean 5-m ocean current velocities. The magenta and blue stars denote the location of the Obidos and Ciudad Bolivar gauge stations, where Amazon and Orinoco river discharges were recorded, respectively. **b** Long-term mean (1979–2018) of observed precipitation averaged over the Amazon river basin in each month. Note that the mean TRMM precipitation is averaged over 1998–2018. **c** Long-term mean of Amazon river discharge (1979–2018) at Obidos and Dai and Trenberth river discharge (1979–2014) in each month. **d** Long-term mean of 5-m ECCO4 (1992–2017), GECCO2 (1979–2016), ORAS5 (1979–2018), SODA3.3.1 (1980–2015), and EN4 (1979–2018) ocean salinity averaged over the APR in each month. The error bars in **b**, **c**, and **d** indicate the standard deviations of each month throughout the analysis period. The geographic map is produced by Python Cartopy package[77].

during the past few decades[24–26] and resulted in increased likelihood of extreme terrestrial events, such as droughts and floods[24,26,27]. Ocean salinity in the APR has also been affected by the seasonality changes in the Amazonia hydroclimatological cycle[4,7,28]. However, it remains unclear if the enhanced precipitation seasonal cycle has intensified the seasonalities of the river discharges and APR ocean salinity by increasing the peaks and deepening the troughs. This study aims to examine the causal link between the changes in the amplitude of the seasonality of the Amazonia hydroclimatological system and APR ocean salinity during the period 1979–2018, using observations, reanalysis, and ocean-state estimate products. The effect of seasonality changes is further quantified using a hierarchical global climate modeling approach.

## Results

**Enhanced seasonalities in observations and reanalysis data**. We first examine the monthly climatological values of precipitation, Amazon river discharge, and APR ocean salinity during the period 1979–2018 (Fig. 1b–d) to illustrate their seasonal cycle linkage. The area-averaged Amazonia precipitation shows the highest values during January–February–March and the lowest during July–August–September (Fig. 1b). Only small differences appear among different observational precipitation datasets (Fig. 1b), indicating the robustness of the estimated annual cycle for the Amazonia precipitation reported in this study.

Following the peak precipitation in March, the Amazon river discharge (expressed as volume transport in m$^3$ s$^{-1}$), observed at Obidos station (magenta star in Fig. 1a), reaches its highest value in June (Fig. 1c). Similarly, the lowest precipitation in August leads to the lowest level of river discharge in November. This 3-month delayed response of the river discharge to precipitation is a prominent hydrological feature in the Amazon river basin[2].

The seasonal cycle of the near-surface (~5 m below the sea surface) ocean salinity in the APR from multiple ocean-state estimate products shows the freshest values in May–June–July and the saltiest in December–January–February (Fig. 1d), which largely follows the seasonal cycle of the Amazon river discharge (Fig. 1c). Although the magnitude of the seasonal cycle varies with the chosen datasets, these ocean-state estimate datasets agree that the APR ocean salinity exhibits a similar seasonal evolution to the Amazon river discharge with no apparent lag. The seasonal cycles of APR ocean near-surface salinity from the 6-year soil moisture and ocean salinity (SMOS, 2011–2016)[29] and 3-year Aquarius (2012–2014)[30] satellite observations have similar characteristics (Supplementary Fig. 1).

The above results suggest that the atmosphere, land, and ocean within and near the Amazon river basin are closely connected through their seasonal cycle characteristics and lag relationships. The annual maximum and minimum values of Amazon precipitation during 1979–2018 indicate that the wet seasons have become wetter and the dry seasons' drier (Fig. 2a). A significant increase in the maximum values and decrease in the minimum values over the analysis period result in overall significant (at 5% level) increasing trends in the precipitation seasonality (Fig. 2b). The increasing trend of the seasonality averaged across the Global Precipitation Climatology Project (GPCP)[31], Global Precipitation Climatology Centre (GPCC)[32], and Precipitation Reconstruction over Land (PREC/L)[33] observational datasets is +0.35 (±0.05) mm day$^{-1}$ decade$^{-1}$. This represents ~6% of the mean seasonality in the Amazon precipitation (~6 mm/day, Fig. 1b). Similar increasing trends of seasonality can be found using Tropical Rainfall Measuring Mission (TRMM)[34] and Climate Prediction Center Merged Analysis of Precipitation (CMAP)[35] datasets (Supplementary Fig. 2a–d).

Similarly, increased seasonality in the Amazon river discharge, due to the increased maximum and decreased minimum values, is also found (Fig. 2c, d). The trend of the increased seasonality in Amazon river discharge is ~1.3 × 10$^4$ m$^3$ s$^{-1}$ decade$^{-1}$, representing about 9% of its mean seasonality (~1.4 × 10$^5$ m$^3$ s$^{-1}$, Fig. 1c). We also examine thirteen other river discharge datasets available within the Amazon river basin (Supplementary Fig. 3), and ten of them show increasing trends throughout the Amazon sub-basins (Supplementary Fig. 4), although the temporal coverages of some river discharge data are too short of providing robust trend estimates (e.g., Supplementary Fig. 4e, i).

Following the enhanced seasonality in the Amazon river discharge, the seasonality trends in APR 5-m ocean salinity (averaged over the red box in Fig. 1a) have also increased by ~2.89 × 10$^{-2}$, 1.82 × 10$^{-2}$, and 1.23 × 10$^{-1}$ PSU decade$^{-1}$ for the German contribution to ECCO version 2 (GECCO2)[36], Estimating the Circulation and Climate of the Ocean project version 4 (ECCO4)[37], and Simple Ocean Data Assimilation version 3.1.1 (SODA3.3.1)[38] products, respectively (Fig. 2f, h). However, we find decreasing trends in the Ocean Reanalysis/analysis version 5 (ORAS5)[39] and EN4[40] products (Fig. 2h). The discrepancy among ocean salinity products is likely related to different data-processing or assimilation procedures, and quality and sampling biases of input data (see discussion in "Methods"). The trend averaged over five products is ~1.44 × 10$^{-2}$ PSU decade$^{-1}$, which accounts for only ~1% of the mean seasonality (~1.38 PSU, Fig. 1d). However, this trend increases to 5.66 × 10$^{-2}$ PSU decade$^{-1}$, ~4% of the mean seasonality, when the three products that show an increasing seasonality trend are averaged (though this estimate is dominated by the increasing trend in the SODA3.3.1 product, red line in Fig. 2h). We also used five gridded ARGO products[41] to determine recent (post-2001) changes in APR ocean salinity seasonality (see Supplementary Fig. 5b); the average of the five available gridded ARGO datasets shows an unambiguous increasing trend (rightmost bar in Supplementary Fig. 5b).

To understand the underlying mechanisms for the seasonal cycle changes, we consider the vertically integrated atmospheric moisture budget (see "Methods"), averaged over the Amazon river basin using multiple reanalysis products. The net increasing trend in precipitation seasonality (Fig. 2b) is found in all the reanalysis datasets (Fig. 3a), which are largely attributed to the enhanced vertical moisture advection (i.e., $-\left\langle \omega \frac{\partial q}{\partial p} \right\rangle$, Fig. 3d) and secondarily to the enhanced horizontal moisture advection (i.e., $-\left\langle \vec{\mathbf{v}} \cdot \nabla q \right\rangle$, Fig. 3c). In contrast, the evapotranspiration (i.e., $E$, Fig. 3b) and residuals (i.e., $\delta$, Fig. 3e) counteract the advection effects. The trend toward increasing seasonality in vertical moisture advection is ~0.52 ± 0.15 mm day$^{-1}$ decade$^{-1}$, which contributes to precipitation of ~0.36 ± 7.15 mm day$^{-1}$ decade$^{-1}$ and is larger than the horizontal moisture advection of 0.16 ± 0.06 mm day$^{-1}$ decade$^{-1}$. Further decomposition of the vertical moisture advection shows that the dynamical component ($-\left\langle \omega' \frac{\partial q}{\partial p} \right\rangle$, 0.45 ± 0.15 mm day$^{-1}$ decade$^{-1}$, Fig. 3g) contributes more strongly than the thermodynamic component ($-\left\langle \omega \frac{\partial q'}{\partial p} \right\rangle$, 0.07 ± 0.03 mm day$^{-1}$ decade$^{-1}$, Fig. 3f). Such increased seasonality in the dynamic component is related to the vertical motion (Fig. 3i) that favors increased seasonality in convective activity above the Amazon river basin, while that in the thermodynamic component is associated with the increasing atmospheric moisture content (Fig. 3j).

The precipitation and evapotranspiration seasonality changes over the APR region should also affect the local ocean salinity. Thus, we perform the same moisture budget analysis over the

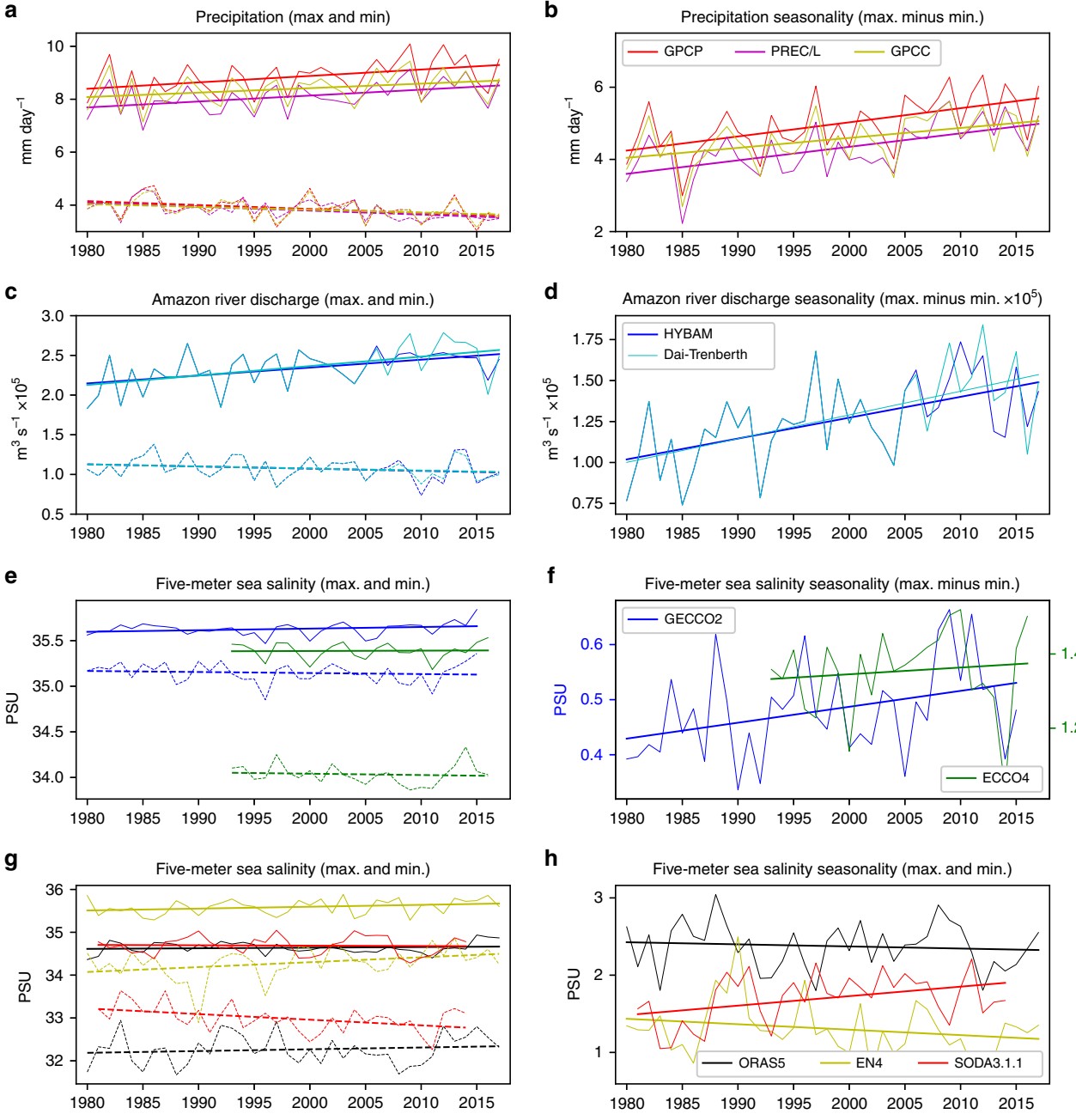

**Fig. 2 Evolution of the annual maximum and minimum values and seasonality for Amazonia precipitation, Amazon river discharge, and Amazon plume region (APR) ocean salinity. a** Amazonia precipitation during the period 1979–2018. **b** The seasonality of Amazonia precipitation (maximum minus minimum values) during the period 1979–2018. **c**, **d**, **e**, **f**, and **g**, **h** are similar to **a**, **b**, but for Amazon river discharge and APR ocean salinity, respectively. Note that ECCO4 salinity data only cover the period 1992–2017; GECCO2 the period 1979–2016; and SODA3.1.3 the period 1980–2015. The solid (dashed) lines are the linear fits used to determine the trends.

APR (red box in Fig. 1a), but find no robust and consistent trends in seasonalities of precipitation and evapotranspiration compared to that of the APR ocean salinity (Supplementary Fig. 6). The precipitation seasonality has a negative trend, while that of evapotranspiration has weakly increased, neither of which can account for the observed robust increasing trend in the APR ocean salinity seasonality. This leaves the increased seasonality in river discharge as a sole source for the observed change in ocean salinity seasonality, although it could be modulated by ocean advection and vertical mixing processes. In addition, the Orinoco river discharge may influence the APR ocean salinity[8]; however, we find its effect is less important than that of the Amazon river discharge, as the

seasonality trend in the Orinoco river discharge has decreased before 2000 and weakly increased ($2.1 \times 10^3$ m$^3$ s$^{-1}$ decade$^{-1}$) after mid-2000 (Supplementary Fig. 7).

**Climate model sensitivity experiments.** We conduct two sets of historical global climate model experiments during 1979–2009 (the availability of the Coordinated Ocean-ice Reference Experiments Phase 2 project, CORE-II, forcing[42] limits the simulation period, see "Methods") to single out the effects of Amazonia precipitation and Amazon river discharge seasonality changes. The experiments also help address the causality. The first set uses

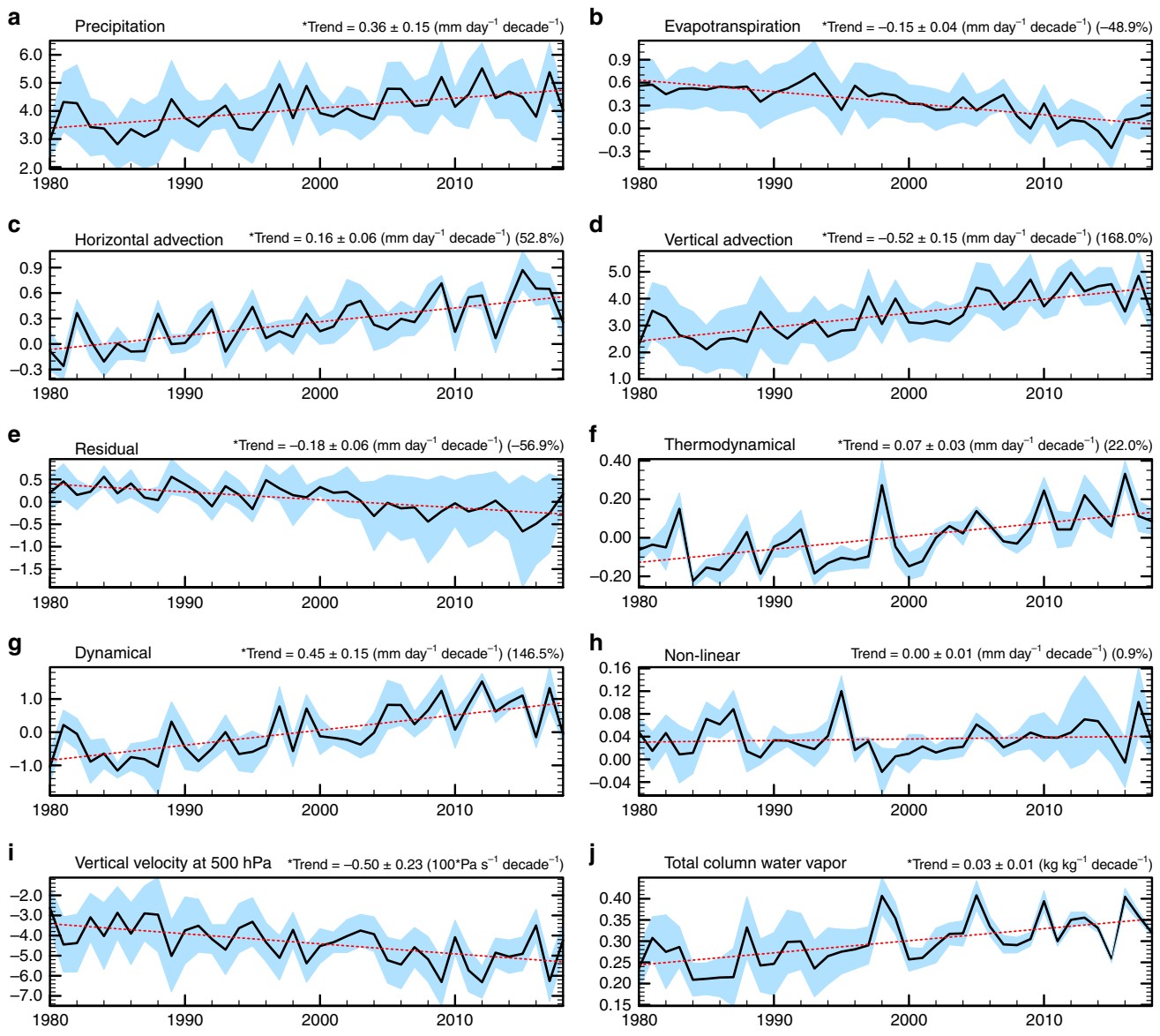

**Fig. 3 Atmospheric moisture budget analysis within the Amazon river basin.** The seasonality of reanalysis precipitation (**a**), evapotranspiration ($E$) (**b**), horizontal moisture advection ($-\langle \vec{v} \cdot \nabla q \rangle$) (**c**), vertical moisture advection ($-\langle \omega \frac{\partial q}{\partial p} \rangle$) (**d**), residual ($\delta$) (**e**), thermodynamic component ($-\langle \bar{\omega} \frac{\partial \bar{q}}{\partial p} \rangle$) (**f**), dynamical component ($-\langle \omega' \frac{\partial \bar{q}}{\partial p} \rangle$) (**g**), nonlinear component ($-\langle \omega' \frac{q'}{\partial p} \rangle$) (**h**), vertical velocity ($\omega$) (**i**), and total moisture ($q$) (**j**) over time. The blue shadings are the range among reanalysis products, and the red lines are the linear fits used to determine trends. Note different $y$-axis ranges.

a global land model with increased seasonality in the precipitation forcing by a factor of 1.5 and 1.75 during the 1979–2009 period over the Amazon river basin (Fig. 4a and see "Methods"). In response to the enhanced precipitation seasonality trend, we find a nearly linear response in the seasonality trend of the Amazon river discharge (Fig. 4b). The relative change in the river discharge seasonality is slightly higher than that in precipitation seasonality; that is, the precipitation seasonality increases by a factor of 1.75, whereas that of the river discharge increases by a factor of 1.87 (Fig. 4b). This runoff intensification effect is likely attributed to nonlinear river discharge responses to precipitation intensity identified in a previous study[27].

To test the robustness of the ocean salinity seasonality response, given the discrepancy shown in the ocean-state estimate products (Fig. 2f, h), we conduct the second set of experiments using a global ocean model forced with varied seasonality in Amazon river discharge (Fig. 4c and see "Methods"). The APR 5-m ocean salinity seasonality (averaged over the red box in Fig. 1a)

also increases nearly linearly as the river discharge seasonality increases (Fig. 4d). It should be noted that the total amount of Amazon freshwater input in each experiment remains the same by our experimental design (see "Methods"); that is, no additional freshwater is added to the model when changing its seasonality. We further analyze 12 CORE-II ocean-only hindcast simulations forced with the same CORE-II forcings, including precipitation and river discharge from 1979 to 2009 (see "Methods"), and obtain overall increased APR ocean salinity seasonality trends (Supplementary Fig. 5a). The above hierarchical historical climate model experiments and CORE-II simulations, therefore, lend support to the increased APR salinity seasonality trends found in GECCO2, ECCO4, and SODA3.1.1 ocean-state estimate products.

The enhanced seasonality in the ocean salinity can affect the ocean physics and dynamics within the APR. Thus, we further examine the trends in the seasonalities of the APR area (Fig. 5a, defined as the area where ocean salinity are less than 34.5 as

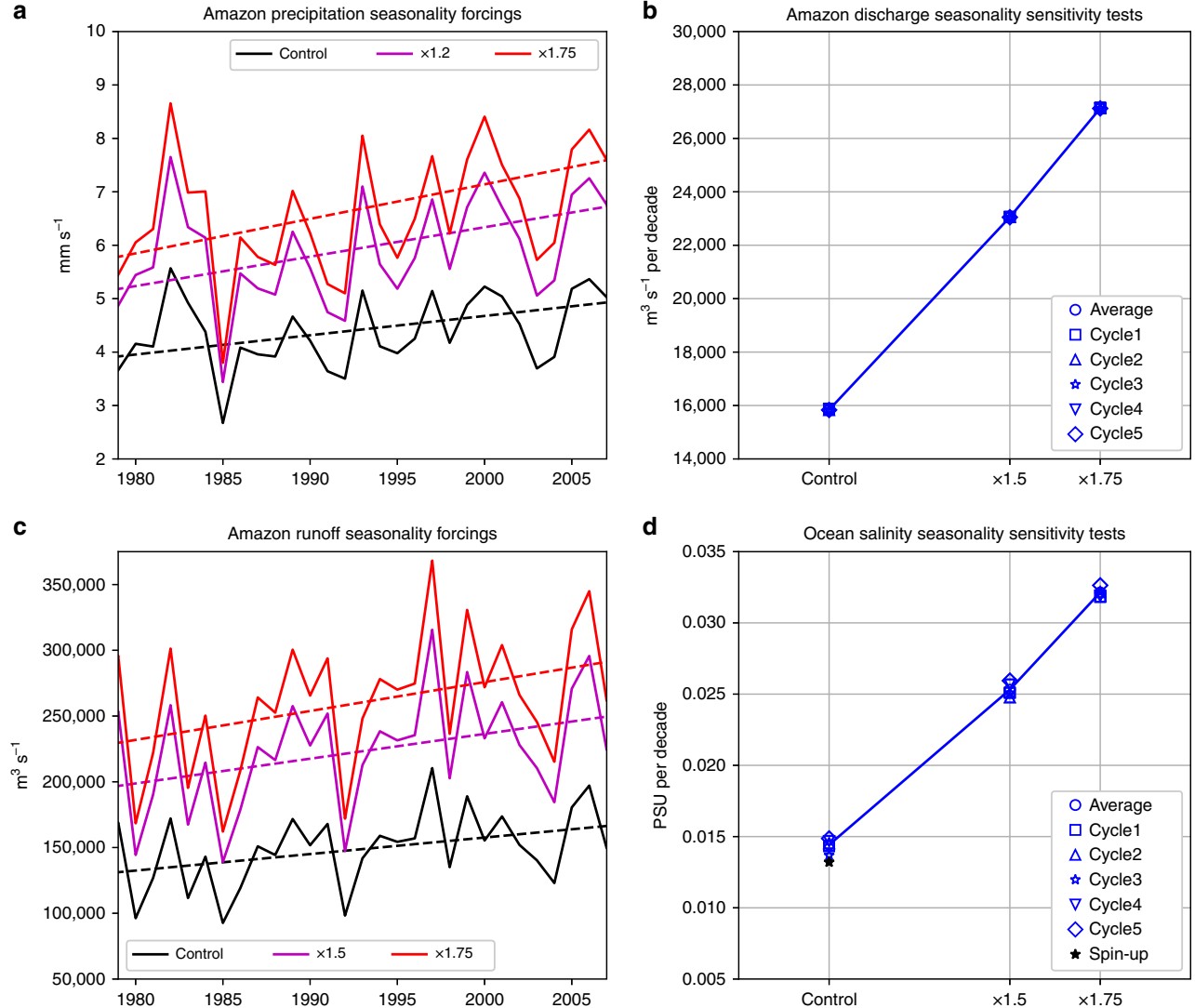

**Fig. 4 Sensitivity of Amazon river discharge and Amazon plume region (APR) ocean salinity in response to seasonality changes in Amazonia precipitation and Amazon river runoff forcings in global land and ocean models. a** The seasonality in the precipitation forcing used in the global land model control and experimental simulations. **b** Seasonality changes in Amazon river discharge in the land model experiments. **c** similar to **a**, but for Amazon river runoff forcing used to force the ocean model experiments. In **a**, **c**, the dashed lines are the linear fits used to determine trends. **d** similar to **b**, but for seasonality changes in APR ocean salinity in the ocean model experiments. The black star is the APR salinity seasonality trend, averaged over five spin-up cycles.

denoted by the magenta contour line in Fig. 1a), the 5-m ocean temperature (Fig. 5b), the upper ocean stratification (i.e., buoyancy frequency, $N^2$, which also indicates the strength of vertical mixing, Fig. 5c), and the barrier-layer thickness (Fig. 5d) from the ocean model experiments. In order to better characterize the localized changes, the averaged values shown in Fig. 5 are taken over only the regions where the 5-m ocean salinity is <34.5 PSU (an alternative definition of APR area) rather than averaged over the APR box as those presented in Fig. 4). However, both the areal metrics produce very similar results (c.f., Supplementary Fig. 8 and Fig. 4d). The seasonality trends of the APR area and vertical mixing strength again respond nearly linearly to that of the Amazon river discharge, whereas the trends of the 5-m ocean temperature and barrier-layer thickness seasonalities show more sensitivity, with larger trend increases from ×1.5 to ×1.75 experiments. These features indicate that the vertical mixing process is dominated by the effect of ocean salinity change associated with the Amazon river freshwater change, manifested as more linear behaviors, but the ocean temperature and barrier-

layer thickness are also affected by other factors. It is possible that the temperature responses are more affected by the variability of internal ocean dynamics (e.g., baroclinic eddies or the bifurcation of the North Brazilian Current)[43,44] manifested as the larger spreads among five ensembles in each experiment (Fig. 5b). In addition, the Amazon river discharge temperature is not accounted for in the ocean model configuration, which may also contribute to the APR ocean temperature response.

## Discussion
This study finds that an amplified seasonal cycle of Amazonia precipitation during the period 1979–2018 leads to enhanced seasonalities in both Amazon river discharge and APR ocean salinity, using a combination of observations and reanalysis datasets. Hierarchical climate model experiments support the observed seasonality changes and shed light on the sole effects of changing seasonalities in the Amazonia precipitation and Amazon river discharge. While previous studies mainly focused on

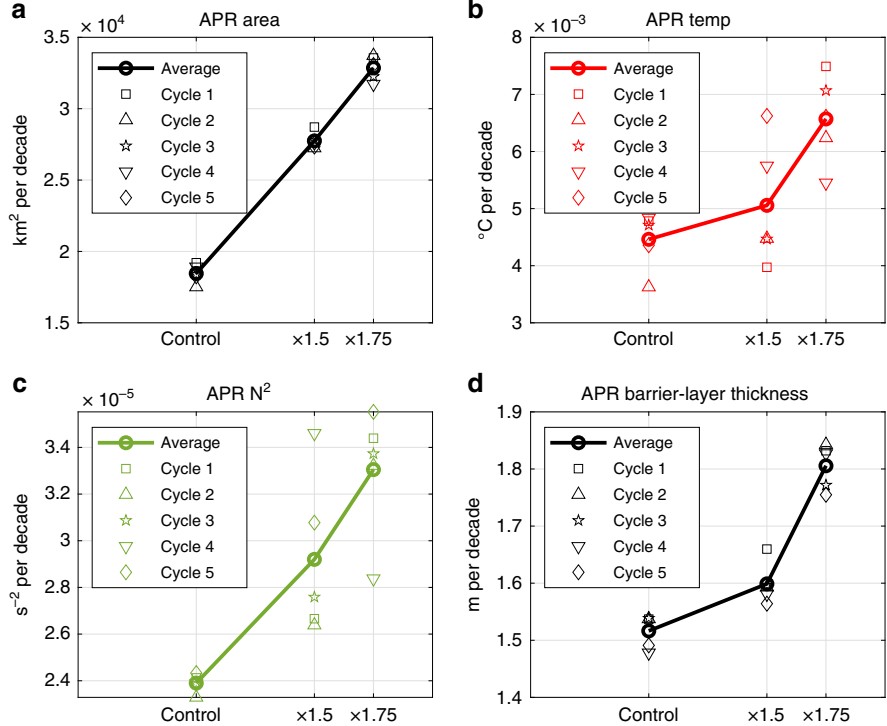

**Fig. 5 Sensitivity of the trends in Amazon plume region (APR) size, ocean temperature, buoyancy frequency ($N^2$), and barrier-layer thickness in response to seasonality changes in Amazon river runoff forcings in ocean model experiments. a** Seasonality changes in the size of APR, defined as the region where 5-m ocean salinity is <34.5 PSU. **b** Seasonality changes in the 5-m ocean temperature averaged over the region where 5-m ocean salinity is <34.5 PSU. **c**, **d** similar to **b**, but for seasonality changes in buoyancy frequency and barrier-layer thickness.

specific dry or wet seasons without taking into account the seasonality changes in a comprehensive fashion[24–26], our results provide a new route to further study the Amazonia hydro-climatology and the occurrence of extreme events in the Amazon river basin and APR.

Within the APR, we find the enhanced seasonality of ocean salinity is tied closely with the enhanced seasonalities in the plume area, upper ocean stratification, near-surface ocean temperature, and barrier-layer thickness in our ocean model experiments (Fig. 5). These changing ocean properties are important in that they could affect the carbon cycle and marine biogeochemistry within the APR more significantly as a consequence of increased seasonality of Amazon river discharge[7]. It is noted that the increasing precipitation trend in the wet season (mostly January–February–March) contributes more than that in the dry season (mostly July–August–September) to the increasing seasonality trend (Fig. 2a), which is also the case for Amazon river discharge (Fig. 2c) and APR ocean salinity (Fig. 2e, g). Presumably, warmer near-surface ocean temperature and thicker barrier layer in the APR in the "fresher" season (mostly May–June–July) could offer favorable surface ocean conditions for hurricane genesis via barrier layer dynamics[4,8,14,15]. Previous studies using statistical and dynamical hurricane forecast framework showed that the inclusion of upper ocean heat content provides longer and better predictability of hurricane intensity[45–47]. The results presented in this study, therefore, have important implications for hurricane forecasting. However, the near-linear relationship in our modeling results does not include the atmosphere–land–ocean feedback processes. There may exist limitations to tie the seasonality changes of Amazon river discharge and APR ocean salinity, and unrealistic seasonality responses. In addition, previous studies have shown that the Amazon river discharge can affect tropical Atlantic air–sea

interactions[17], regional sea-level height[20–22], and have potentially far-reaching impacts on the AMOC[23].

The atmospheric moisture budget analysis reveals that the "wet-get-wetter-and-dry-get-drier" phenomenon in the tropical precipitation due to anthropogenic global warming[48] may contribute to the seasonality change. The "wet-get-wetter-and-dry-get-drier" precipitation signal can result in a host of consistent seasonality responses in the Amazonia atmosphere–land–ocean coupled system. The enhanced vertical velocity seasonality (Fig. 3i) may reflect changes in the location and strength of the Atlantic ITCZ that are related to changes in the local sea-surface temperature gradient in the tropical Atlantic[49].

It should be noted that the decadal and multidecadal natural modes of variability, such as Pacific decadal variability (PDV), interdecadal Pacific variability (IPV), or Atlantic multidecadal variability (AMV), are possible large-scale drivers of the precipitation seasonality changes at longer timescales[50,51], which have been shown by some studies to be more influential than anthropogenic forcing in the Amazon river basin in observations and climate model simulations[52,53]. For example, recent dry season droughts across the Amazon river basin have been attributed to the AMV[54,55]. We find that the AMV, PDV, and IPV indices and the seasonality of the Amazon river discharge time series, after applying 11-year running average, are correlated after 1970 ($R = 0.58$ for AMV index, $R = -0.81$ for PDV index, $R = -0.64$ for IPV, all of which are larger than 0.41, critical value at 99% significance level, see Supplementary Fig. 9), indeed suggesting that part of the increased Amazon river discharge seasonality trend in the past 30 years can be attributed to low-frequency Atlantic and Pacific sea-surface temperature variations. In addition, changes in hydropower dam construction[56], deforestation[57], and groundwater dynamics[58] may have also affected the hydroclimatology of Amazonia and consequently river

discharge seasonality. All these effects are not considered in this study. To quantify the relative and combined effect of all local versus remote forcings and natural versus anthropogenic factors would require significant modeling efforts, including a series of well-designed global climate model experiments. This will be left for a future study.

## Methods

**APR and the Amazon river basin**. The APR is chosen according to the extent to which the effect of Amazon river freshwater can reach in terms of overall mean state[8]. Previous studies showed that the size of the freshwater plume is determined by the combination of the region of relatively low salinity extending from the Amazon river mouth (magenta contour line in Fig. 1a) and strength of the prevailing North Brazilian Current (denoted by black arrows near coasts in Fig. 1a); it carries the freshwater released at the Amazon river mouth northwestward to the Caribbean[4,7], though sometimes the North Brazilian Current turns eastward to bring freshwater eastward[43]. The APR we used in this study (red box in Fig. 1a) covers not only the near-coastal area but also this bifurcation branch of ocean circulation. The APR is used to calculate the area-averaged near-surface ocean salinity time series. We also consider the region with ocean salinity <34.5 PSU following a previous study[8] to define APR (magenta contour line in Fig. 1a), which produces similar results (c.f., Fig. 4d and Supplementary Fig. 8).

We define the Amazon river basin (black contour line in Fig. 1a) as the catchment upstream of the Obidos station based on ArcGISV10.1, which is different from the conventional Amazon river basin based on the Amazon river mouth near the Equator (black contour in Supplementary Fig. 3). Because the conventional definition includes river discharge downstream of the Obidos station, this adjusted river basin is used to more accurately calculate the area-averaged Amazonia precipitation time series consistent with river discharge at the Obidos station. We compare area-averaged precipitation annual cycles using this adjusted and conventional river basin (black contour line in Supplementary Figs. 3 and 10a). Slightly less precipitation using the adjusted river basin occurs in the wet season, and more precipitation from spring to fall. We also compare the absolute difference ratio between them and find the ratio ranges from 2.5% to 21% (Supplementary Fig. 10b). The correlation coefficient between their monthly time series during 1979–2018 is as high as 0.99. These results using the adjusted river basin largely capture the amount and variability of those using the conventional Amazon river basin.

**The observational, reanalysis, and ocean-state estimate datasets**. The monthly Amazon river discharge observed at Obidos gauge station (#17050001, 1.9225°S, and 55.6753°W, magenta star in Fig. 1a) for 1968–2018 and the Orinoco river discharge data observed at Ciudad Bolivar gauge station (#408000000, 08.1536°N, and 063.5361°W, blue star in Fig. 1a) for 2003–2018 are obtained from SO HYBAM material transport datasets (formerly Environmental Research Observatory, http://www.ore-hybam.org/). Other river discharge data within the Amazon river basin are also obtained from the HYBAM website (Supplementary Figs. 3 and 4). We also use the Amazon river and Orinoco river discharge data from the Global River Flow and Continental Discharge Data Set during 1979–2018 (http://www.cgd.ucar.edu/cas/catalog/surface/dai-runoff/)[59], which provides river discharge data for the world's 925 largest rivers primarily based on gauge observations with the assistance of model simulations.

For the observational monthly precipitation datasets, we use GPCP version 6 (https://www.esrl.noaa.gov/psd/data/gridded/data.gpcp.html)[31,60], GPCC (https://www.esrl.noaa.gov/psd/data/gridded/data.gpcp.html)[32], and Precipitation Reconstruction over Land (PREC/L, https://www.esrl.noaa.gov/psd/data/gridded/data.precl.html)[33] for 1979–2019. We also use observational precipitation datasets from the TRMM version 7 (https://pmm.nasa.gov/data-access/downloads/trmm)[34] and the Climate Prediction CMAP (https://www.esrl.noaa.gov/psd/data/gridded/data.cmap.html)[35]. Multiple reanalysis products during 1979–2018 are used when calculating the atmospheric water moisture budget (see Supplementary Tables 1 and 4–8). We have compared the variability of reanalysis precipitation datasets with the observed ones. Their characteristics are very similar (c.f., Figs. 2b and 3a).

When calculating the effect of evapotranspiration[61,62], we use multiple reanalysis products (Supplementary Table 4). We also use global monthly evapotranspiration fields from the Global Land Evaporation Amsterdam Model (GLEAM, during 1980–2018, https://www.gleam.eu/)[63,64], which assimilates a series of land surface and satellite observations. Their results are largely similar (c.f. Fig. 3b and Supplementary Fig. 2f).

Several monthly observational and ocean-state estimate products for the salinity field at the surface and 5 m below the ocean surface used in this study include the SMOS (during 2011–2016, https://www.esa.int/Our_Activities/Observing_the_Earth/SMOS)[29] and the U.S./Argentina Aquarius/SACD (during 2012–2014, https://aquarius.oceansciences.org/cgi/index-noflash.htm)[30], Estimating the Circulation and Climate of the Ocean project version 4 (ECCO4, during 1992–2017, www.ecco-group.org)[37] and the German contribution to ECCO version 2 (GECCO2, during 1979–2016, https://icdc.cen.uni-hamburg.de/1/daten/reanalysis-ocean/gecco2.html)[36], EN4.2.1 (EN4 hereafter, during 1979–2018, https://www.metoffice.gov.uk/hadobs/en4/en4-0-2-profile-file-format.html)[40],

Ocean Reanalysis/analysis version 5 (ORAS5, during 1979–2018, https://www.ecmwf.int/en/research/climate-reanalysis/ocean-reanalysis)[39], and Simple Ocean Data Assimilation version 3.1.1 (SODA3.1.1, during 1980–2015, http://www.soda.umd.edu/)[38]. We also use five interpolated ARGO products (see the first four bars from the left in Supplementary Fig. 5b, http://www.argo.ucsd.edu/Gridded_fields.html)[41].

We examine the data quality of EN4 according to its salinity observation weight and uncertainty in the APR (Supplementary Fig. 11). Despite the increasing observation weight and decreasing salinity uncertainty after the late 1990s in the APR; low observation weight and large uncertainty before the mid-1990s may contribute to the discrepancy of APR salinity seasonality between EN4 and other products. In addition, different assimilation process in generating the products may also be a factor.

Before calculating the area-averaged values, we only regrid the field from ORAS5 and GECCO2 product to ~1° × 1° using nearest interpolation because the former is output in T-grid and the latitude grid of the latter varies in different variables. We do not perform interpolation for other datasets.

**Atmospheric moisture budget analysis**. We utilize a vertically integrated moisture budget analysis to explore the mechanisms behind the enhanced precipitation seasonality. A similar analysis has been performed in many studies to examine global and regional precipitation changes[27,65] on various timescales (i.e., daily, monthly, and interannually). The moisture budget is formulated as

$$P = -\left\langle \omega \frac{\partial q}{\partial p} \right\rangle - \langle \vec{v} \cdot \nabla q \rangle + E + \delta, \qquad (1)$$

where $P$ denotes precipitation, $\vec{v}$ the horizontal velocity field, $q$ specific humidity, $E$ evapotranspiration, $\delta$ the residual term, $\omega$ is the pressure velocity, and <> mass integration throughout the atmospheric layers (surface to model top). The first term on the right-hand side $\left(-\left\langle \omega \frac{\partial q}{\partial p} \right\rangle\right)$ represents the vertical moisture advection, while the second term $(-\langle \vec{v} \cdot \nabla q \rangle)$ represents the horizontal moisture advection $(-\langle \vec{v} \cdot \nabla q \rangle)$. When vertical integration is performed on $-\left\langle \omega \frac{\partial q}{\partial p} \right\rangle$, the pressure velocities at the surface and at the model top are assumed to be zero. Note that the residual term ($\delta$) includes transient eddy and nonlinear effects.

The vertical moisture advection can be further divided into:

$$-\left\langle \omega \frac{\partial q}{\partial p} \right\rangle' \cong -\left\langle \bar{\omega} \frac{\partial q'}{\partial p} \right\rangle - \left\langle \omega' \frac{\partial \bar{q}}{\partial p} \right\rangle, \qquad (2)$$

where $\overline{()}$ indicates seasonal averaging from 1980 to 2018 and $()'$ denotes the seasonal anomaly from the seasonal mean in the wet and dry seasons. We disregard the nonlinear term $-\left\langle \omega' \frac{\partial q'}{\partial p} \right\rangle$. This decomposition allows us to examine the dynamical and thermodynamical contributions to precipitation changes. The first term of the above equation on the right-hand side represents the thermodynamical term, while the second term, the dynamical term respectively follows previous studies[65–67]. The unit of each term is kg s$^{-1}$ m$^{-2}$, which is equivalent to ml s$^{-1}$.

**Seasonality calculation and enhancement**. Since this study focuses on the seasonal averages of precipitation, river discharge, and APR ocean salinity, we take 3-month averages before calculating their seasonality. The seasonality of the time series with 3-monthly averaging is defined as the difference of its maximum value minus its minimum value within 1 year in this study. Similar results can be obtained without taking a 3-month average. We also calculate seasonality with the difference between fixed wet and dry seasons (e.g., January–February–March averaged precipitation minus July–August–September averaged precipitation based on the climatological seasonal cycle, Fig. 1b), and obtain very similar results.

To enhance the seasonality for a given forcing field in climate model experiments, we use a fast Fourier transformation (FFT) approach. We first apply FFT on a target time series, and before applying inverse FFT to retrieve the resultant time series, we multiply a targeted factor to enhance its amplitude. To double the amplitude, for example, we choose the factor as 2. To demonstrate, we consider a simple combined sine wave, $\sin\left(\frac{2\pi}{360}x\right) + \sin\left(\frac{4\pi}{360}x\right) + 5$, which is shown as blue line in Supplementary Fig. 12, while the time series that has been amplified by a factor of 2 is shown as the red line. It is noted that the mean of the two time series is exactly the same.

**Global land model historical experiments**. To test the sensitivity and quantify the seasonality of Amazon river discharge changes in response to Amazonia precipitation seasonality changes, we use the Community Land Model version 4.5 (CLM4.5)[63] under the Community Earth System Model (CESM) framework to conduct land-only experiments with varying precipitation forcing seasonality. The atmospheric conditions used to force the CLM4.5 hindcast experiments are constructed following a previous study[64] using observational and reanalysis datasets from 1948 to 2009. For the control simulation, we conduct a 62-year simulation from 1948 to 2009 with corresponding forcings prescribed. We then repeat four cycles with the same forcings to generate a total of five ensemble members. For the "x1.5" ("x1.75") experiment, we conduct another 62-year simulation from 1948 to 2009 cycling five times using the precipitation forcing seasonality increased by a

factor of 1.5 (1.75) above the Amazon river basin. We only analyze the results from 1979 to 2009.

When we construct precipitation forcing with enhanced seasonality, due to the fact that precipitation forcing is given in 6-h time intervals, we first take the monthly average from the 6-h precipitation field and then perform FFT on the monthly field to enhance the seasonality. We then add increased or decreased values in 1 month back to the original 6-hourly precipitation forcing for the experimental simulations. However, some resultant 6-h values can be less than zero, which is not reasonable, so we set all negative values to zero. Although in this way, the mean of the resultant precipitation forcing is not exactly the same as that of the original precipitation forcing, we compare their mean values and find only a small difference.

**Global ocean model historical experiments**. We conduct an ocean-only experiment similar to the land-only experiment described above, but with the river runoff forcing seasonality changed at the Amazon river mouth, to examine the sensitivity of ocean salinity in the APR. We use the Parallel Ocean Program version 2 (POP2) under the CESM framework. The boundary conditions used to force POP2 are prepared according to the Coordinated Ocean-ice Reference Experiments Phase 2 project (CORE-II)[42], which spans from 1948 to 2009. In the global river runoff forcing to drive the ocean model, we increase the river runoff at the grid, where largest annual mean river runoff occurs in the South American continent, by a factor of 1.5 and 1.75 to construct runoff forcings for the "x1.5" experiment and the "x1.75" experiment, respectively, using the FFT approach as well. It is noted that the global river runoff forcing is constructed based on the Dai-Trenberth's dataset[59], which is analyzed in Figs. 1 and 2.

Due to the fact that the ocean model requires a longer time to reach quasi-equilibrium and to effectively reduce model drift in the historical ocean-only simulations, a five-cycle spin-up simulation was suggested by previous studies[68,69]. Therefore, we conduct five-cycle spin-up simulations with repeating boundary conditions from 1948 to 2009 (black star in Fig. 4d). The control simulation is continued from the spin-up run for another five more cycles to generate five ensemble members, whereas the "x1.5" and the "x1.75" experiments are continued from the spin-up runs for another five more cycles given river runoff forcings with Amazon river runoff forcing seasonalities increased by a factor of 1.5 and 1.75, respectively. We only analyze the results from 1979 to 2009.

The original runoff and amplified seasonalities of river runoff are shown in Fig. 4c. We choose the amplification factors in ocean model experiments in order to prevent the amplified runoff value from being smaller than zero. It is noted that the total river runoff released into the ocean is exactly the same in each experiment because the monthly mean of runoff fields is the same, which is a direct result of the FFT approach described above.

**Coordinated Ocean-ice Reference Experiments Phase 2**. Coordinated Ocean-ice Reference Experiments Phase 2 (CORE-II) entails a set of coordinated historical global ocean model experiments using different state-of-the-art global ocean models developed by different modeling groups. The models are prescribed with common forcings, including precipitation and river runoff from 1948 to 2009[42,70]. The boundary fluxes are computed following the same bulk formulae[42]. CORE-II simulations provide a framework to investigate mechanisms of significant ocean phenomena and their seasonal and decadal variabilities, including both forced and internal variability. Therefore, CORE-II fits well for this study to examine whether the increasing seasonality trend of near-surface ocean salinity in the APR is a common feature in response to the observed precipitation and subsequent river runoff seasonality changes in the Amazon river basin.

Supplementary fig. 5a shows that the near-surface ocean seasonalities in 12 CORE-II simulations are overall enhanced during 1979–2007 (we drop the last two years because some models do not provide simulated results in 2008 and 2009), which is consistent with the result found in the GECCO4, ECCO, and SODA3.1.1 ocean-state estimate products. The multi-model mean trend is about 0.054 PSU per decade. The values of the increased seasonality trend are comparable to those in the ocean-state estimate products (~0.04 PSU per decade for GECCO2 and ~0.03 PSU per decade for ECCO4), indicating that CORE-II simulations reasonably capture the increased trends of ocean salinity seasonality and support our findings based on ocean-state estimate products. For a comparison of interannual variability between simulated long-term mean seasonal cycles, one is referring to a previous study[42]. CORE-II simulation results are downloaded from the NCAR/NCEP Research Data Archive (https://rda.ucar.edu/datasets/ds262.0/).

**Natural variability indices**. To assess potential linkages between the seasonality in the Amazonia hydroclimatological system and natural variability, we consider the AMV[71], PDV[72], and IPV[73], as shown in Supplementary Fig. 9. The time series are shown as 11-year running average to illustrate their decadal variability. The AMV index and a tripole index representing IPV are downloaded from ESRL Physical Science Division (https://www.esrl.noaa.gov/psd/data/timeseries/AMO/, and https://www.esrl.noaa.gov/psd/data/timeseries/IPOTPI/), and the PDV index is obtained from the Joint Institute for the Study of the Atmosphere and Ocean (http://research.jisao.washington.edu/pdo/).

**Near-surface ocean mixing and barrier-layer calculations**. In order to examine the responses of near-surface ocean physics and dynamics to Amazon river discharge in the APR, we calculate the barrier-layer thickness and potential energy following a previous study[74]. The barrier-layer thickness is defined as the isothermal layer depth (ILD) minus the mixed layer depth (MLD) when the former is deeper than the latter. If ILD is shallower than MLD, the barrier-layer thickness at this grid is not considered. The MLD is calculated as $\sigma_{ref} + \Delta\sigma$, where $\sigma_{ref}$ is chosen as 5 m and $\Delta\sigma$ 0.1 kg m$^{-3}$. The ILD is computed using the temperature difference equivalence to 0.1 kg m$^{-3}$ of density increase from the reference depth with the salinity at the reference 5-m depth. We only consider grid points where the 5-m ocean salinity is <35.4 PSU within the APR to better characterize the freshwater plume following a previous study[8], but similar results can be obtained without this constraint. To characterize the near-surface mixing processes, we also calculate the squared buoyancy frequency (in the unit of s$^{-2}$), defined as:

$$N^2 = -\frac{g}{\rho_0}\frac{\partial\rho}{\partial z},$$

where $g$ is the gravitational constant, $\rho_0$ is a reference density (1025 kg m$^{-3}$), $\rho$ is density, and $z$ is depth.

**Statistical significance test**. For a given time series, the statistical significance of its trend is determined based on a Student's $t$ test with a null hypothesis that the trend is zero[75]. If the $P$ value is <0.05, the null hypothesis can be rejected with 5% significance, and the trend is considered significant at the 5% level. We consider the effective sample size when performing the $t$ test to take into account the effect of serial correlation. The effective sample size (ESS) is given as:

$$\text{ESS} = N\frac{1 - R_x R_y}{1 + R_x R_y},$$

where $N$ is the length of time series, and $R_x$ and $R_y$ are the lag-1 autocorrelations of time series $x$ and $y$, respectively[76].

## Data availability

The monthly Amazon river discharge observed at Obidos gauge station and the Orinoco river discharge data observed at Ciudad Bolivar gauge station are obtained from SO HYBAM material transport datasets (formerly Environmental Research Observatory, http://www.ore-hybam.org/). We also use the Amazon river and Orinoco river discharge data from the Global River Flow and Continental Discharge Data Set (http://www.cgd.ucar.edu/cas/catalog/surface/dai-runoff/). For the observational monthly precipitation datasets, we use Global Precipitation Climatology Project version 6 (https://www.esrl.noaa.gov/psd/data/gridded/data.gpcp.html), Global Precipitation Climatology Centre (https://www.esrl.noaa.gov/psd/data/gridded/data.gpcc.html), and Precipitation Reconstruction over Land (https://www.esrl.noaa.gov/psd/data/gridded/data.precl.html). We also use observational precipitation datasets from the TRMM version 7 (https://pmm.nasa.gov/data-access/downloads/trmm) and the Climate Prediction Center Merged Analysis of Precipitation (https://www.esrl.noaa.gov/psd/data/gridded/data.cmap.html). Multiple reanalysis datasets are used: ERAI is obtained from ECMWF (https://www.ecmwf.int/en/forecasts/datasets/reanalysis-datasets/era-interim), MERRA from NASA (https://gmao.gsfc.nasa.gov/reanalysis/MERRA/), JRA and JRA-55 from the Japan Meteorological Agency and the Central Research Institute of Electric Power Industry (https://jra.kishou.go.jp/JRA-55/index_en.html), NCEP_R2 from ESRL (https://www.esrl.noaa.gov/psd/data/gridded/data.ncep.reanalysis2.html), and NCEP_CFSR from NCAR/UCAR (https://climatedataguide.ucar.edu/climate-data/climate-forecast-system-reanalysis-cfsr). The Global Land Evaporation Amsterdam Model data are downloaded from https://www.gleam.eu/. Several monthly observational and ocean-state estimate products for the salinity field at the surface and 5 m below the ocean surface used in this study include the soil moisture and ocean salinity (https://www.esa.int/Our_Activities/Observing_the_Earth/SMOS) and the U.S./Argentina Aquarius/SACD (https://aquarius.oceansciences.org/cgi/index-noflash.htm), Estimating the Circulation and Climate of the Ocean project version 4 (www.ecco-group.org) and the German contribution to ECCO version 2 (https://icdc.cen.uni-hamburg.de/1/daten/reanalysis-ocean/gecco2.html), EN4.0.2 (https://www.metoffice.gov.uk/hadobs/en4/en4-0-2-profile-file-format.html), Ocean Reanalysis/analysis version 5 (https://www.ecmwf.int/en/research/climate-reanalysis/ocean-reanalysis), and Simple Ocean Data Assimilation version 3.1.1 (http://www.soda.umd.edu/). We also use five interpolated ARGO products (http://www.argo.ucsd.edu/Gridded_fields.html). Ocean-only simulations of Coordinated Ocean-ice Reference Experiments Phase 2 are obtained from NCAR/UCAR (https://rda.ucar.edu/datasets/ds262.0/). The AMV index and a tripole index are downloaded from ESRL Physical Science Division (https://www.esrl.noaa.gov/psd/data/timeseries/AMO/, and https://www.esrl.noaa.gov/psd/data/timeseries/IPOTPI/), and the PDV index is obtained from the Joint Institute for the Study of the Atmosphere and Ocean (http://research.jisao.washington.edu/pdo/). CORE-II simulation results are downloaded from the NCAR/NCEP Research Data Archive (https://rda.ucar.edu/datasets/ds262.0/). ERAI and ERA5 datasets are downloaded from ECMWF (https://www.ecmwf.int/en/forecasts/datasets/reanalysis-datasets/era-interim and https://www.ecmwf.int/en/forecasts/datasets/reanalysis-datasets/era5). JRA-55 data are downloaded from JRA Project (https://jra.kishou.go.jp/JRA-55/index_en.html#download). NCEP_R1 and NCEP_R2 are downloaded from NOAA's Physical Sciences Laboratory (https://psl.noaa.gov/data/

gridded/index.html). The sensitivity climate model simulations are compiled on the Zenodo data repository (https://doi.org/10.5281/zenodo.3939611). The geographic maps in Fig. 1a and Supplementary Fig. 3 are produced by Python Cartopy package (https://scitools.org.uk/cartopy/docs/latest/#)[77].

## Code availability

The codes that analyze the data and make figures are available on Y.-C. L.'s GitHub website (https://github.com/yuchiaol/Amazon_river_seasonality).

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

## Acknowledgements

M.-H.L., C.-W.L., and R.-J.W. are supported by the Ministry of Science and Technology in Taiwan under grant 106-2111-M-002-010-MY4. H.S. and J.D.S. are grateful for support from NOAA NA19OAR4310376 and NA17OAR4310255. C.C.U. acknowledges support from the U.S. National Science Foundation under grant OCE-1663704. The National Center for Atmospheric Research (NCAR) is a major facility sponsored by the US National Science Foundation (NSF) under Cooperative Agreement No. 1852977. We thank Dr. Young-Oh Kwon at Woods Hole Oceanographic Institution and Dr. Who Kim at NCAR for discussions about the ocean model experiment design. We thank Dr. Mehnaz Rashid at National Taiwan University and Wen-Yin Wu at the University of Texas at Austin in helping generate the high-resolution Amazon river mask. We also thank Dr. Gael Forget at Massachusetts Institue of Technology for comments on using ECCO and other ocean-state estimate products.

## Author contributions

The paper was conceived and written by Y.-C.L. and M.-H.L. with inputs from C.-W.L., H.S., C.C.U., S.Y., R.-J.W., and J.D.S. The global ocean and land model experiments were conducted by Y.-C.L. and R.-J.W. C.-W.L. performed the atmospheric moisture budget analysis. J.D.S. performed the analyses on surface ocean properties and barrier-layer thickness.

## Competing interests

The authors declare no competing interests.
