## [Peer Review File · Nature Communications]

Reviewers' comments:

Reviewer #1 (Remarks to the Author):

Liang et al. presented here the time series of the seasonality changes in Amazonia precipitation, Amazon river discharge and ocean salinity of the Amazon plume region. They found all the three seasonalities have been enhanced during the period 1979-2014. The authors draw links between the observations, that is, amplified precipitation seasonality results in enhanced seasonality in river discharge, which subsequently leads to increased seasonality in APR ocean salinity. They also examined the mechanism that has intensified the seasonal change in Amazonia precipitation. Furthermore, Liang et al. performed both global land and ocean models to test the relationships between these changes in seasonalities.

I found that overall, the paper is well written and easy to follow. I therefore support its publication if the authors can satisfyingly address the following concerns in the revision.

Major concerns:

1. The freshwater distribution in the Amazon plume region can change seasonally due to the strength of trade winds and seasonal shift of the ITCZ. So, the freshwater is a mixture of Amazon river waters from different seasons. The salinity minimum can even appear before the maximum river discharge. See Coles et al. (2013) which is also cited by the authors. If so, there must be some limitations to tie the seasonal change in the APR ocean salinity to the seasonality in Amazon river discharge.
2. More explanations are needed for the model simulations on seasonality changes. There are discrepancies between the control results (Fig. 4a, c) and the observations (Fig. 2b, d). It is probably not realistic that the seasonality in precipitation and runoff could increase by 1.5 to 1.75 times. For example, the observed seasonality increase is 5% or less (lines 154-159). Then, how robust can the sensitivity tests be used to quantify the relationship between their seasonality changes?

Minor comments:

3. Lines 144-146. In supplementary figure 4, the increasing trend in seasonality of river discharge only appears in 6 out of the total 13 rivers.
4. Lines 159-162. Supplementary figure 5 shows 6, not 5 interpolated ARGO products.
5. Lines 178-181. How does the vertical motion of moisture cause the increased seasonality? Figure 3i shows a decreasing trend in vertical velocity at 500 hPa. Can the authors elaborate the mechanism here?
6. Lines 231-233. "Fresher" instead of "saltier" season? Also, "boreal summer" instead of "summer"?
7. Line 244, "Fig. 3i", not "Fig. 2i".

Reviewer #2 (Remarks to the Author):

Amplified Seasonal Cycle in Hydroclimate over the Amazon River Basin and its Plume Region in the Tropical Atlantic

The authors connect the amplification of the Amazon River basin seasonal cycle to the amplification of the Amazon Plume Region salinity. The paper is very well written and contributes substantial knowledge towards our understanding of the global impacts of changes in the Amazon River basin hydro-climatology.

It merits publication in Nature Communications but I would like the authors to address the following points before reaching a final conclusion:

1) One important aspect of this paper is the connection between found increased seasonality in the Amazon hydro-climate and global impacts. For instance: in line 81-83 the authors mention how APR salinity impacts hurricane genesis, carbon cycle and climate variability.

This for me is the most important aspect of the results and a brief description (a sentence) of how, for instance, decreased salinity in the APR impacts hurricane genesis would greatly improve the narrative. Same with the carbon cycle impacts and Atlantic climate variability. If these connections are clearer from the beginning, the reader can better grasp the impact of the results.

2) Why does the analysis end in 2014? Would a trend be significant if the El Niño wet season drought of 2015-2016 is included in the series? The 3 datasets presented in the results of Fig.2 have updated data through 2019 and Obidos discharge data through 2018. Same for reanalysis: why does reanalysis data end in 2010?

The authors should make an effort to update their analysis to the most updated time series possible as the results discuss long-term trends and multi-decadal modes of variability.

3) The authors mention in lines 246-252 the relationship between multi-decadal modes of variability and responses in the Amazon hydro-climate. This deserves more attention as PDV and AMV have been shown to impact the region (see references below and please cite) more significantly than long term trends, and patterns currently observed are susceptible to change along with phase changes in PDV and AMV as shown in supplementary Fig. 9.

Marengo, J. A. (2009). Long-term trends and cycles in the hydrometeorology of the Amazon basin since the late 1920s. *Hydrological Processes: An International Journal*, 23(22), 3236-3244

Fernandes, K., Giannini, A., Verchot, L., Baethgen, W., & Pinedo-Vasquez, M. (2015). Decadal covariability of Atlantic SSTs and western Amazon dry-season hydroclimate in observations and CMIP5 simulations. *Geophysical Research Letters*, 42(16), 6793-6801.

4) How is the seasonality calculated in the reanalysis? In observations it is the difference between the maximum and minimum seasonal precipitation within a year. Is the reanalysis seasonality calculated based on the observed annual max and min, or the reanalysis max and min? Could they represent annual amplitudes based on different seasons? Models (even reanalysis) do not necessarily reproduce observed seasonal cycles.

5) Related to point #1: In line 208 the justification for using a global ocean model seems to be to verify observations, which is counter intuitive. However, in lines 223-227 the authors briefly mention the impact that modeled salinity increased seasonality has on stratification, temperature in barrier layer thickness in the APR. My suggestion is that this result, which better justifies the use of model simulations, is more thoroughly discussed and Supplemental Fig.8 becomes part of the main text.

Minor adjustments:

1) I suggest units in Fig.3 are converted to mm/day for closer comparison. Reanalysis is not expected to perfectly reproduce observed precipitation amounts, but trends and differences (such as annual precipitation amplitude) can be better interpreted that way.

2) Line 229-230: Sentence: "It is noted that the increasing precipitation trend in the wet season contributes more to the increasing seasonality trend compared to decreasing trend in the dry season, which is also the case for Amazon river discharge in the wet season...". Note that the trend in maximum river discharge does not occur during the wet season (as defined by precipitation). I suggest the authors refer to the season being discussed by the months it is composed of (ex: DJF,

MJJ, etc). Same for "summer" reference in Line 233.

3) Lines 231-233: "Warmer near-surface ocean temperature and thicker barrier layer in the APR in the 'saltier' season (mostly summer) could offer favourable conditions for hurricane genesis".

My understanding is that barrier layer thickness is associated with a layer of fresh water (less dense) at the ocean surface? In other words, shouldn't thicker barrier layer occur in the "fresher/highest river discharge rate" rather than "saltier" season?

4) Please carefully review the text for typos.

Reviewers' comments:

Reviewer #1 (Remarks to the Author):

Liang et al. presented here the time series of the seasonality changes in Amazonia precipitation, Amazon river discharge and ocean salinity of the Amazon plume region. They found all the three seasonalities have been enhanced during the period 1979-2014. The authors draw links between the observations, that is, amplified precipitation seasonality results in enhanced seasonality in river discharge, which subsequently leads to increased seasonality in APR ocean salinity. They also examined the mechanism that has intensified the seasonal change in Amazonia precipitation. Furthermore, Liang et al. performed both global land and ocean models to test the relationships between these changes in seasonalities.

I found that overall, the paper is well written and easy to follow. I therefore support its publication if the authors can satisfyingly address the following concerns in the revision.

Reply: We appreciate the reviewer's support. Please find below our point-by-point replies to the reviewer's comments.

Major concerns:

1. The freshwater distribution in the Amazon plume region can change seasonally due to the strength of trade winds and seasonal shift of the ITCZ. So, the freshwater is a mixture of Amazon river waters from different seasons. The salinity minimum can even appear before the maximum river discharge. See Coles et al. (2013) which is also cited by the authors. If so, there must be some limitations to tie the seasonal change in the APR ocean salinity to the seasonality in Amazon river discharge.

Reply: The strength of the tropical Atlantic trade winds and seasonal shift of the Atlantic ITCZ indeed can influence the freshwater distribution in the Amazon plume region (APR), and we admit some limitations exist. Based on our analysis on the vertical wind averaged over the APR region (Supplementary Fig. 6j), we find a weak, insignificant trend of vertical velocity. If the ITCZ or trade winds in the tropical Atlantic, especially those that reach the APR, play significant roles in affecting the freshwater input, one would expect that changes in vertical velocity over the APR emerge as a robust signal. However, this does not appear to be the case in our results. More comprehensive analyses on the trade wind and ITCZ changes are needed to better quantify their effects, but this is beyond the current study and could be taken up in a separate future study. Additionally, the ocean model experiments conducted in this study, in which atmospheric wind forcings are imposed identically, are designed to isolate the effect of Amazon river freshwater independent of changes in trade winds or ITCZ. It would be informative to also quantify the effects of trade wind and ITCZ by, for example, wind forcing relaxation method (considering a lead-lag relationship) in another set of ocean model or fully-coupled experiments. In this study, we intend to focus on the APR salinity change following, not in advance of, the Amazon river discharge. We have added relevant discussions in the revised manuscript to this effect (Lines 291-294).

2. More explanations are needed for the model simulations on seasonality changes. There are discrepancies between the control results (Fig. 4a, c) and the observations (Fig. 2b, d). It is probably not realistic that the seasonality in precipitation and runoff could increase by 1.5 to 1.75 times. For example, the observed seasonality increase is 5% or less (lines 154-159). Then, how robust can the sensitivity tests be used to quantify the relationship between their seasonality changes.

Reply: The motivation to conduct the ocean model sensitivity experiments with a factor of 1.5 and 1.75 is actually based on the abrupt rise (or drop) of Amazon river discharge seasonality that was observed in the past: For example, $\sim 1 \times 10^5 \text{ m}^3/\text{s}$ of river discharge seasonality in 2004 increased to $\sim 1.4 \times 10^5 \text{ m}^3/\text{s}$ in 2005,

leading to an increase of approximately 40% (if considering the increase from 2004 to 2006, a rise of approximately 60% occurred). Therefore, seasonality increases of 1.5 or 1.75 times may correspond to some extreme cases, though we are focusing on the changes in trends of seasonality. Thus, our simulations could provide insight towards observed seasonality changes. In this study, some processes tied to the air-sea interaction may not be realistically simulated because we conducted ocean-only experiments without including dynamical atmosphere and associated atmosphere-ocean coupling processes. Therefore, the seasonal differences between the control simulations and observations that the reviewer pointed out could occur. This deserves further studies using fully-coupled simulations with Amazon river discharge perturbed. We have added relevant discussions in the revised manuscript (Lines 291-295).

Minor comments:

3. Lines 144-146. In supplementary figure 4, the increasing trend in seasonality of river discharge only appears in 6 out of the total 13 rivers.

Reply: We now include the trends in the title of each panel and only 3 sub-basins show decreasing trends (Supplementary Figs. 4a, d, and i). As such, 10 out of 13 rivers show increasing trends, although some of the trends are not significant.

4. Lines 159-162. Supplementary figure 5 shows 6, not 5 interpolated ARGO products.

Reply: In Supplementary Fig. 5, the right-most bar indicates the average of the 5 products. We have modified the sentence for clarification in the revised manuscript (Lines 175-176; 408-409).

5. Lines 178-181. How does the vertical motion of moisture cause the increased seasonality? Figure 3i shows a decreasing trend in vertical velocity at 500 hPa. Can the authors elaborate the mechanism here?

Reply: We used “omega” to represent vertical motion, and the decreasing trend of omega indicated an increasing trend of upward air motion. As there is a trend towards more ascent, the convective activity and precipitation in the Amazon river basin are expected to be higher. We have added discussions in the revised manuscript (Lines 192-193).

6. Lines 231-233. “Fresher” instead of “saltier” season? Also, “boreal summer” instead of “summer”

Reply: We thank the reviewer’s correction and suggestion, and have modified the text accordingly in the revised manuscript (Lines 286-288).

7. Line 244, “Fig. 3i”, not “Fig. 2i”.

Reply: We have corrected this typo in the revised manuscript (Line 304).

Reviewer #2 (Remarks to the Author):

Amplified Seasonal Cycle in Hydroclimate over the Amazon River Basin and its Plume Region in the Tropical Atlantic

The authors connect the amplification of the Amazon River basin seasonal cycle to the amplification of the Amazon Plume Region salinity. The paper is very well written and contributes substantial knowledge towards our understanding of the global impacts of changes in the Amazon River basin hydro-climatology.

It merits publication in Nature Communications but I would like the authors to address the following points before reaching a final conclusion:

Reply: We thank the reviewer for their support. Please find below our point-by-point replies to the reviewer's comments.

1) One important aspect of this paper is the connection between found increased seasonality in the Amazon hydro-climate and global impacts. For instance: in line 81-83 the authors mention how APR salinity impacts hurricane genesis, carbon cycle and climate variability.

This for me is the most important aspect of the results and a brief description (a sentence) of how, for instance, decreased salinity in the APR impacts hurricane genesis would greatly improve the narrative. Same with the carbon cycle impacts and Atlantic climate variability. If these connections are clearer from the beginning, the reader can better grasp the impact of the results.

Reply: We thank the reviewer for their suggestions, and have now elaborated our descriptions of the effects of salinity changes on hurricane genesis, carbon cycle, and Atlantic climate variability in the introductory paragraphs (Lines 76-95). For the marine biogeochemistry, productivity, and carbon cycle in the APR, we have referenced previous observational studies, which showed that the high nutrient content of Amazon river discharge sustains high marine-productivity in the APR with maximum chlorophyll content concentrated in the upper 5-meter ocean (e.g., Smith Jr and Demaster, 1996), as well as the mixture of supersaturated Amazon freshwater and undersaturated surface ocean water results in a sink of atmospheric carbon dioxide within the APR (e.g., Körtzinger, 2003; Ibáñez et al. 2015) (Lines 76-82). For hurricane genesis, we have mentioned that the low salinity water in the APR is known to produce thick barrier layers via modulating upper ocean stratification, which enables the ocean to store more heat in the surface layers/upper ocean (e.g., Field, 2007; Vizzy and Cook, 2010; Grodsky et al., 2012; Coles et al., 2013; Grodsky et al., 2014) (Lines 83-91). The additional heat storage in the APR barrier layer provides favourable surface ocean conditions for hurricane genesis. For Atlantic climate, we have provided a brief overview that the variability of Amazon freshwater and resultant ocean salinity changes have been suggested to affect tropical Atlantic air-sea interactions (Masson and Delecluse, 2001; Rudzin et al. 2019), the variability of the Atlantic intertropical convergence zone (ITCZ, Jahfer et al., 2020), and potentially far-reaching impacts on the Atlantic meridional overturning circulation (AMOC, Jahfer et al., 2017) (Lines 91-95). We have also mentioned recent studies about possible impacts of Amazon river discharge on regional sea-level height (Durand et al., 2019; Giffard et al., 2019; Piecuch and Wadehra, 2020) (Lines 93-94). Finally, to enhance the applicability of our results on hurricane and marine biogeochemistry, we have revised the original Supplementary Fig. 8 as the new Fig. 5 and added discussions in the revised manuscript (Lines 240-263; 265-298).

2) Why does the analysis end in 2014? Would a trend be significant if the El Niño wet season drought of 2015-2016 is included in the series? The 3 datasets presented in the results of Fig.2 have updated data through 2019 and Obidos discharge data through 2018. Same for reanalysis: why does reanalysis data end in 2010?

The authors should make an effort to update their analysis to the most updated time series possible as the results discuss long-term trends and multi-decadal modes of variability.

Reply: We have extended the observational and analysis periods to December of 2018 (or as long as possible, for example, GECCO2 data till December of 2016, ECCO4 December of 2017, SODA3.3.1 December of 2015) in the revised Figs. 2-3 and Supplementary Figs. 2, 6, 7, 9, and 10. The results are largely unchanged and the trends are still significant even after including the 2015-2016 El Niño period.

3) The authors mention in lines 246-252 the relationship between multi-decadal modes of variability and responses in the Amazon hydro-climate. This deserves more attention as PDV and AMV have been shown to impact the region (see references below and please cite) more significantly than long term trends, and patterns currently observed are susceptible to change along with phase changes in PDV and AMV as shown in supplementary Fig. 9.

Marengo, J. A. (2009). Long-term trends and cycles in the hydrometeorology of the Amazon basin since the late 1920s. *Hydrological Processes: An International Journal*, 23(22), 3236-3244

Fernandes, K., Giannini, A., Verchot, L., Baethgen, W., & Pinedo-Vasquez, M. (2015). Decadal covariability of Atlantic SSTs and western Amazon dry-season hydroclimate in observations and CMIP5 simulations. *Geophysical Research Letters*, 42(16), 6793-6801.

Reply: We have added relevant discussions about the roles of PDV, IPV, and AMV together with the insights from the two references offered by the reviewer in the revised manuscript (Lines 307-319). First of all, we discuss that these modes of climate variability could be more influential than the global warming trend effect over the Amazon river basin as suggested by Marengo (2009) and Fernandes et al. (2015) (Lines 310-320). Then we refer to two previous studies that focused on how AMV affects hydroclimate over the Amazon river basin (Lewis et al., 2011; Marengo et al., 2012) (Lines 312-313). Lastly, we calculate correlation coefficients between the Amazon river discharge time series and PDV, IPV, AMV indices to inform and quantify their potential relationship (Lines 314-318 and supplementary Fig. 9).

4) How is the seasonality calculated in the reanalysis? In observations it is the difference between the maximum and minimum seasonal precipitation within a year. Is the reanalysis seasonality calculated based on the observed annual max and min, or the reanalysis max and min? Could they represent annual amplitudes based on different seasons? Models (even reanalysis) do not necessarily reproduce observed seasonal cycles.

Reply: The seasonalities calculated in Fig. 3 are based on maximum and minimum values (after a 3-month running average was performed on the monthly time series, following the method by Chou et al., 2013) in the reanalysis precipitation. Indeed, there exist some discrepancies between reanalysis and observational datasets, as we can see the differences in terms of the amplitude and variability (e.g., Supplementary Figs. 6a and b). However, their trends show consistent features, and the variabilities are largely similar (e.g., peak values in 1988 and 2010, and low values in 1989 and 2000), although some discrepancies in the mean state of their seasonal amplitude exist. Therefore, the analyses based on reanalysis datasets should be able to represent important features and evolutions of annual amplitude found in observations at least in the interannual timescale. We also compare the mean seasonal cycles of Amazon river discharge from the land model control experiments and the APR ocean salinity from the ocean model control experiments to those from observational and reanalysis datasets (c.f. Figs. 1c-d, and Fig. R1). Their seasonal evolutions are largely similar.

Fig. R1 | Mean seasonal cycle of Amazon river discharge (i.e., river runoff in land model output at the grid closest to Obidos station) from the land model control experiments (a) and APR ocean salinity from the ocean model control experiments (b).

5) Related to point #1: In line 208 the justification for using a global ocean model seems to be to verify observations, which is counter intuitive. However, in lines 223-227 the authors briefly mention the impact that modeled salinity increased seasonality has on stratification, temperature in barrier layer thickness in the APR. My suggestion is that this result, which better justifies the use of model simulations, is more thoroughly discussed and Supplemental Fig.8 becomes part of the main text.

Reply: We thank the reviewer for this suggestion and have now moved the original Supplementary Fig. 8 to new Fig. 5 in the revised manuscript. In the revised manuscript, we discuss results shown in Fig. 5, including the seasonality changes in the size of the Amazon plume (strictly defined by the regions with lower salinity than 34.5 PSU following Ffield, 2007), 5-meter ocean temperature, buoyancy frequency, and barrier layer thickness (Lines 240-263). Then we discuss how these surface ocean properties changes could affect marine biogeochemistry and hurricane forecast (Lines 276-298), echoing what we have reviewed in the introductory paragraphs (Lines 76-99).

Minor adjustments:

1) I suggest units in Fig.3 are converted to mm/day for closer comparison. Reanalysis is not expected to perfectly reproduce observed precipitation amounts, but trends and differences (such as annual precipitation amplitude) can be better interpreted that way.

Reply: We have changed the units in Fig. 3 and Supplementary Fig. 6 to mm/day in the revised manuscript. The unit conversion information is described in the Method section of the revised manuscript (Lines 446-447).

2) Line 229-230: Sentence: “It is noted that the increasing precipitation trend in the wet season contributes more to the increasing seasonality trend compared to decreasing trend in the dry season, which is also the case for Amazon river discharge in the wet season...”. Note that the trend in maximum river discharge does not occur during the wet season (as defined by precipitation). I suggest the authors refer to the season being discussed by the months it is composed of (ex: DJF, MJJ, etc). Same for “summer” reference in Line 233.

Reply: We have made these modifications, now referring to months in the revised manuscript (Lines 279-288).

3) Lines 231-233: “Warmer near-surface ocean temperature and thicker barrier layer in the APR in the ‘saltier’ season (mostly summer) could offer favourable conditions for hurricane genesis”.

My understanding is that barrier layer thickness is associated with a layer of fresh water (less dense) at the ocean surface? In other words, shouldn't thicker barrier layer occur in the "fresher/highest river discharge rate" rather than “saltier” season?

Reply: We thank the reviewer for their correction and have revised the sentence in the revised manuscript (Lines 281-288).

4) Please carefully review the text for typos.

Reply: We have carefully reviewed the wordings in the revised manuscript and corrected all typos.

References

Coles, V. J., Brooks, M. T., Hopkins, J., Stukel, M. R., Yager, P. L., & Hood, R. R. The pathways and properties of the Amazon River Plume in the tropical North Atlantic Ocean. *J. Geophys. Res.: Oceans* **118**, 6894-6913 (2013).

Chou, C., Chiang, J. C., Lan, C. W., Chung, C. H., Liao, Y. C., & Lee, C. J. Increase in the range between wet and dry season precipitation. *Nat. Geosci.* **6**, 263 (2013).

Durand, F., Picuch, C. G., Becker, M., Papa, F., Raju, S. V., Khan, J. U., & Ponte, R. M. Impact of Continental Freshwater Runoff on Coastal Sea Level. *Surv. Geophys.* **40**, 1437-1466 (2019).

Fernandes, K., Giannini, A., Verchot, L., Baethgen, W. & Pinedo-Vasquez, M. Decadal covariability of Atlantic SSTs and western Amazon dry-season hydroclimate in observations and CMIP5 simulations. *Geophys. Res. Lett.* **42**, 6793–6801 (2015).

Ffield, A. Amazon and Orinoco River plumes and NBC rings: bystanders or participants in hurricane events? *J. Clim.* **20**, 316-333 (2007).

Giffard, P., Llovel, W., Jouanno, J., Morvan, G., & Decharme, B. Contribution of the Amazon River Discharge to Regional Sea Level in the Tropical Atlantic Ocean. *Water* **11**, 2348 (2019).

Grodsky, S. A., et al. Haline hurricane wake in the Amazon/Orinoco plume: AQUARIUS/SACD and SMOS observations. *Geophys. Res. Lett.* **39** (2012).

Grodsky, S. A., Reverdin, G., Carton, J. A., & Coles, V. J. Year-to-year salinity changes in the Amazon plume: Contrasting 2011 and 2012 Aquarius/SACD and SMOS satellite data. *Remote Sens. Environ.* **140**, 14-22 (2014).

Ibáñez, J. S. P., Diverrès, D., Araujo, M., & Lefèvre, N. Seasonal and interannual variability of sea-air CO₂ fluxes in the tropical Atlantic affected by the Amazon River plume. *Global Biogeochem. Cy.* **29**, 1640-1655 (2015).

Jahfer, S., Vinayachandran, P. N., & Nanjundiah, R. S. Long-term impact of Amazon river runoff on northern hemispheric climate. *Sci. Rep.* **7**, 10989 (2017).

- Jahfer, S., Vinayachandran, P. N., & Nanjundiah, R. S. The role of Amazon river runoff on the multidecadal variability of Atlantic ITCZ. *Environ. Res. Lett.* (2020).
- Körtzinger, A. A significant CO₂ sink in the tropical Atlantic Ocean associated with the Amazon River plume. *Geophys. Res. Lett.* **30** (2003).
- Lewis, S. L., Brando, P. M., Phillips, O. L., van der Heijden, G. M. F., & Nepstad, D. The 2010 Amazon drought. *Science* **331**, 554 (2011).
- Marengo, J. A. Long-term trends and cycles in the hydrometeorology of the Amazon basin since the late 1920s. *Hydrol. Processes* **23**, 3236-3244 (2009).
- Marengo, J. A., Tomasella, J., Soares, W. R., Alves, L. M., Nobre, C. A. Extreme climatic events in the Amazon basin. *Theor. Appl. Climatol.* **107**, 73-85 (2012).
- Masson, S., & Delecluse, P. Influence of the Amazon river runoff on the tropical Atlantic. *Physics and Chemistry of the Earth, Part B: Hydrology, Oceans and Atmosphere*, **26**, 137-142 (2001).
- Piecuch, C. G., & Wadehra, R. Dynamic Sea Level Variability due to Seasonal River Discharge: A Preliminary Global Ocean Model Study. *Geophys. Res. Lett.*, **47**, e2020GL086984 (2020).
- Smith Jr, W. O., & Demaster, D. J. Phytoplankton biomass and productivity in the Amazon River plume: correlation with seasonal river discharge. *Cont. Shelf Res.* **16**, 291-319 (1996).
- Vizy, E. K., & Cook, K. H. (2010). Influence of the Amazon/Orinoco Plume on the summertime Atlantic climate. *J. Geophys. Res.: Atmospheres* 115 (2010).

REVIEWERS' COMMENTS:

Reviewer #1 (Remarks to the Author):

Liang et al. have carefully addressed my comments. I am overall satisfied with the reply and can recommend accepting the manuscript. However, there are quite a few typos and grammar errors in the revision, particularly in those revised sentences. They need to be corrected to meet the high publication standard of Nature Communications.

Some are listed here:

- 1, Line 43, what's "its"? Better to be clear: its hydroclimate variations ..."
- 2, Line 45, "... potential far-reaching influences ..."
- 3, Line 45, "Atlantic climate" is very vague. Better to say "climate over the tropical Atlantic".
- 4, Lines 97-98, this sentence needs to reword. Forecast local marine ecosystem?
- 5, Line 101, Better to say "... seasonality changes in ..."
- 6, Line 107, "weakening the troughs"? Better to say "deepening the troughs"
- 7, Line 158, "... temporal coverages of ..."
- 8, line 167, "... the trend ..."
- 9, line 169, "... this trend increases to ..."
- 10, line 201, "... which can account ..."
- 11, lines 205-208, grammar errors
- 12, lines 211-215, reword this sentence.
- 13, line 220, I would suggest to say "slightly higher" instead of "more apparent than", as it is not obvious from the figure, nor from the difference in factor between 1.75 and 1.87 if considering model discrepancies.
- 14, lines 261-263, What about "Additionally, Amazon river discharge temperature is not accounted for ..."?
- 15, line 290, delete "information"
- 16, line 333, relatively low ...
- 17, line 420, the former is ...
- 18, line 463, we choose the factor ...
- 19, line 470, reword the sentence.
- 20, line 495, but with the river runoff ...
- 21, line 509, we conduct ...
- 22, line 565, deeper than the latter ...

Supplementary Information:

- 23, lines 180-181, while the black denotes those ...

Reviewer #2 (Remarks to the Author):

The authors did a very good job at addressing the Reviewers' comments and I recommend the manuscript for publication.

REVIEWERS' COMMENTS:

Reviewer #1 (Remarks to the Author):

Liang et al. have carefully addressed my comments. I am overall satisfied with the reply and can recommend accepting the manuscript. However, there are quite a few typos and grammar errors in the revision, particularly in those revised sentences. They need to be corrected to meet the high publication standard of Nature Communications.

Reply: We appreciate reviewer's efforts in improving the presentation of this work. We are glad that reviewer is satisfied with our replies and supports publication of our manuscript on Nature Communications. We have revised all the sentences reviewer highlighted and made further corrections on typos and grammar errors in the revised manuscript.

Some are listed here:

- 1, Line 43, what's "its"? Better to be clear: its hydroclimate variations ..."
- 2, Line 45, "... potential far-reaching influences ..."
- 3, Line 45, "Atlantic climate" is very vague. Better to say "climate over the tropical Atlantic".
- 4, Lines 97-98, this sentence needs to reword. Forecast local marine ecosystem?
- 5, Line 101, Better to say "... seasonality changes in ..."
- 6, Line 107, "weakening the troughs"? Better to say "deepening the troughs"
- 7, Line 158, "... temporal coverages of ..."
- 8, line 167, "... the trend ..."
- 9, line 169, "... this trend increases to ..."
- 10, line 201, "... which can account ..."
- 11, lines 205-208, grammar errors
- 12, lines 211-215, reword this sentence.
- 13, line 220, I would suggest to say "slightly higher" instead of "more apparent than", as it is not obvious from the figure, nor from the difference in factor between 1.75 and 1.87 if considering model discrepancies.
- 14, lines 261-263, What about "Additionally, Amazon river discharge temperature is not accounted for ..."?
- 15, line 290, delete "information"
- 16, line 333, relatively low ...
- 17, line 420, the former is ...
- 18, line 463, we choose the factor ...
- 19, line 470, reword the sentence.
- 20, line 495, but with the river runoff ...
- 21, line 509, we conduct ...
- 22, line 565, deeper than the latter ...

Supplementary Information:

- 23, lines 180-181, while the black denotes those ...

Reviewer #2 (Remarks to the Author):

The authors did a very good job at addressing the Reviewers' comments and I recommend the manuscript for publication.

Reply: We appreciate reviewer's efforts in improving the presentation of this work. We are glad that reviewer is satisfied with our replies and recommends publication of our manuscript on Nature Communications.